# Incidence and prevalence of primary care antidepressant prescribing in children and young people in England, 1998–2017: A population-based cohort study

Ruth H. Jack[1]*, Chris Hollis[2,3,4], Carol Coupland[1], Richard Morriss[2,3,4,5], Roger David Knaggs[6], Debbie Butler[4], Andrea Cipriani[7], Samuele Cortese[2,3,4,8,9,10,11], Julia Hippisley-Cox[12]

1 Division of Primary Care, School of Medicine, University of Nottingham, Nottingham, United Kingdom, 2 Division of Psychiatry and Applied Psychology, School of Medicine, University of Nottingham, Nottingham, United Kingdom, 3 National Institute for Health Research (NIHR) Nottingham Biomedical Research Centre, Nottingham, United Kingdom, 4 NIHR MindTech MedTech Co-operative, Nottingham, United Kingdom, 5 NIHR Applied Research Collaboration East Midlands, Nottingham, United Kingdom, 6 School of Pharmacy, University of Nottingham, Nottingham, United Kingdom, 7 Department of Psychiatry, University of Oxford, Oxford, UK; Oxford Health NHS Foundation Trust, Warneford Hospital, Oxford, United Kingdom, 8 Centre for Innovation in Mental Health, School of Psychology, Life and Environmental Sciences, University of Southampton, Southampton, United Kingdom, 9 Clinical and Experimental Sciences (CNS and Psychiatry), Faculty of Medicine, University of Southampton, Southampton, United Kingdom, 10 Solent NHS Trust, Southampton, United Kingdom, 11 New York University Child Study Center, New York, New York, United States of America, 12 Nuffield Department of Primary Care Health Sciences, University of Oxford, Oxford, United Kingdom

* ruth.jack@nottingham.ac.uk

## Abstract

### Background

The use of antidepressants in children and adolescents remains controversial. We examined trends over time and variation in antidepressant prescribing in children and young people in England and whether the drugs prescribed reflected UK licensing and guidelines.

### Methods and findings

QResearch is a primary care database containing anonymised healthcare records of over 32 million patients from more than 1,500 general practices across the UK. All eligible children and young people aged 5–17 years in 1998–2017 from QResearch were included. Incidence and prevalence rates of antidepressant prescriptions in each year were calculated overall, for 4 antidepressant classes (selective serotonin reuptake inhibitors [SSRIs], tricyclic and related antidepressants [TCAs], serotonin and norepinephrine reuptake inhibitors [SNRIs], and other antidepressants), and for individual drugs. Adjusted trends over time and differences by social deprivation, region, and ethnicity were examined using Poisson regression, taking clustering within general practitioner (GP) practices into account using multilevel modelling. Of the 4.3 million children and young people in the cohort, 49,434 (1.1%) were prescribed antidepressants for the first time during 20 million years of follow-up. Males

**Data Availability Statement:** The patient-level data from the QResearch database are specifically

licensed according to its governance framework. Data access is limited to researchers who meet the eligibility criteria and have their project approved by the QResearch Scientific Committee. See https://www.qresearch.org/information/information-for-researchers/ for more details, including how to apply for data access.

**Funding:** This research was funded by the National Institute for Health Research (NIHR) Nottingham Biomedical Research Centre (grant number IS-BRC-1215-20003) https://www.nihr.ac.uk/explore-nihr/support/experimental-medicine.htm, and conducted by the NIHR Nottingham Biomedical Research Centre in collaboration with the Oxford Health Biomedical Research Centre (grant number IS-BRC-1215-20005). RM is also supported by the NIHR MindTech MedTech and In Vitro Diagnostic Co-operative https://www.nihr.ac.uk/partners-and-industry/industry/access-to-expertise/medtech.htm, and the NIHR Applied Research Collaboration East Midlands https://www.nihr.ac.uk/explore-nihr/support/collaborating-in-applied-health-research.htm. AC is supported by the NIHR Oxford Cognitive Health Clinical Research Facility, by an NIHR Research Professorship (grant number RP-2017-08-ST2-006) https://www.nihr.ac.uk/explore-nihr/academy-programmes/research-professorships.htm, by the NIHR Oxford and Thames Valley Applied Research Collaboration https://www.arc-oxtv.nihr.ac.uk, and by the NIHR Oxford Health Biomedical Research Centre (grant number IS-BRC-1215-20005) https://www.nihr.ac.uk/explore-nihr/support/experimental-medicine.htm. The funders had no role in study design, data collection and analysis, decision to publish, or preparation of the manuscript.

**Competing interests:** I have read the journal's policy and the authors of this manuscript have the following competing interests: CC has received a research grant from Nottingham BRC. RDK is a member of the Advisory Council on the Misuse of Drugs and Guideline Committee member, NICE Clinical Guideline on Chronic Pain. AC has received research and consultancy fees from INCiPiT (Italian Network for Paediatric Trials), CARIPLO Foundation, and Angelini Pharma. JHC is professor of clinical epidemiology and general practice at the University of Oxford and co-director of QResearch, a not-for-profit organisation that is a joint partnership between the University of Oxford and Egton Medical Information Systems (the leading commercial supplier of IT for 60% of general practices in the UK). JHC was also a paid director of ClinRisk Ltd, which produces open- and closed-source software to ensure the reliable and updatable implementation of clinical risk equations

made up 52.0% of the cohorts, but only 34.1% of those who were first prescribed an antidepressant in the study period. The largest proportion of the cohort was from London (24.4%), and whilst ethnicity information was missing for 39.5% of the cohort, of those with known ethnicity, 75.3% were White. Overall, SSRIs (62.6%) were the most commonly prescribed first antidepressant, followed by TCAs (35.7%). Incident antidepressant prescribing decreased in 5- to 11-year-olds from a peak of 0.9 in females and 1.6 in males in 1999 to less than 0.2 per 1,000 for both sexes in 2017, but incidence rates more than doubled in 12- to 17-year-olds between 2005 and 2017 to 9.7 (females) and 4.2 (males) per 1,000 person-years. The lowest prescription incidence rates were in London, and the highest were in the South East of England (excluding London) for all sex and age groups. Those living in more deprived areas were more likely to be prescribed antidepressants after adjusting for region. The strongest trend was seen in 12- to 17-year-old females (adjusted incidence rate ratio [aIRR] 1.12, 95% confidence interval [95% CI] 1.11–1.13, $p < 0.001$, per deprivation quintile increase). Prescribing rates were highest in White and lowest in Black adolescents (aIRR 0.32, 95% CI 0.29–0.36, $p < 0.001$ [females]; aIRR 0.32, 95% CI 0.27–0.38, $p < 0.001$ [males]). The 5 most commonly prescribed antidepressants were either licensed in the UK for use in children and young people (CYP) or included in national guidelines. Limitations of the study are that, because we did not have access to secondary care prescribing information, we may be underestimating the prevalence and misidentifying the first antidepressant prescription. We could not assess whether antidepressants were dispensed or taken.

## Conclusions

Our analysis provides evidence of a continuing rise of antidepressant prescribing in adolescents aged 12–17 years since 2005, driven by SSRI prescriptions, but a decrease in children aged 5–11 years. The variation in prescribing by deprivation, region, and ethnicity could represent inequities. Future research should examine whether prescribing trends and variation are due to true differences in need and risk factors, access to diagnosis or treatment, prescribing behaviour, or young people's help-seeking behaviour.

## Author summary

### Why was this study done?

- A substantial increase has been noted in the UK and other countries in the prescription of antidepressant medicines for young people, despite their benefits and safety remaining matters for debate.

- Several antidepressants are included in UK guidelines for children and young people, though not all are currently licensed for use in under 18s in the UK.

- It remains unclear whether there is variation in prescribing across different groups and if the choice of medicines used in real-world practice adheres to evidence-based clinical guidelines.

within clinical computer systems to help improve patient care.

**Abbreviations:** ADHD, attention-deficit hyperactive disorder; aIRR, adjusted IRR; CAMHS, Child and Adolescent Mental Health Services; CI, confidence interval; CYP, children and young people; EMIS, Egton Medical Information Systems; GP, general practitioner; HES, hospital episode statistics; IRR, incidence rate ratio; MAOI, monoamine oxidase inhibitor; NHS, National Health Service; NICE, National Institute for Health and Care Excellence; NIHR, National Institute for Health Research; SNRI, serotonin and norepinephrine reuptake inhibitor; SSRI, selective serotonin reuptake inhibitor; TCA, tricyclic and related antidepressant.

## What did the researchers do and find?

- We examined the incidence and prevalence of antidepressant prescribing in a cohort of over 4.3 million 5- to 17-year-olds in England from a large primary care database between 1998 and 2017.

- Antidepressant prescribing decreased in 5- to 11-year-olds between 1999 and 2017 but more than doubled in 12- to 17-year-olds between 2005 and 2017.

- There was variation in prescribing by deprivation, region, and ethnicity, even after taking other factors into account.

## What do these findings mean?

- The variation in prescribing by deprivation, region, and ethnicity found could represent inequities in care and service provision.

- Future research should examine whether prescribing trends and variation are due to differences in need or barriers in accessing diagnosis or treatment.

## Introduction

Depressive disorders were the third largest cause of adolescent disability-adjusted life years lost globally in 2015 [1]. Compared with adults, children and young people (CYP) with major depressive disorder are still underdiagnosed and undertreated [2,3]. Consequences of depressive episodes in young people include serious impairments in social functioning and school performance, as well as suicidal ideation and attempts [4]. Psychological treatments are still considered the first-line treatment in many clinical guidelines, including the UK National Institute for Health and Care Excellence (NICE) guidelines for depression in CYP [5]. However, 22.7% of CYP with emotional disorders reported waiting more than 6 months to see a mental health specialist in England in 2017 [6], and antidepressants are widely used in the treatment of depression in children and adolescents.

The efficacy and safety of antidepressant medicines for major depression in CYP remains controversial [7]. Fluoxetine is the only antidepressant licensed for use in CYP as a first-line treatment for major depression in the UK [8] and the US [9]. In the UK, other antidepressants are licensed for obsessive-compulsive disorder (fluvoxamine and sertraline) and nocturnal enuresis (imipramine), and these drugs are recommended as the first antidepressants to use by the relevant NICE guidelines [10,11]. Amitriptyline has neuropathic pain listed as an unlicensed indication in the British National Formulary for Children [8] and is suggested as a prophylactic treatment for migraines in NICE guidelines on headaches in over-12s [12]. Tricyclic antidepressants, in particular imipramine, have historically been used as a second or third line drug in the management of attention-deficit hyperactivity disorder (ADHD) in children despite being unlicensed for this indication [8]. Therefore, examining the trends in antidepressants prescribed to CYP may indicate changes in relation to the indications these drugs are licensed for, as well as whether NICE guidelines and evidence-based practice are being followed. According to NICE guidelines, in the UK, prescribing antidepressants to CYP should

only be done after assessment and diagnosis by a child and adolescent psychiatrist [5,10] or other specialist with expertise in child and adolescent mental health [11]. Referral to specialists in secondary care would usually be made, when appropriate, after visiting a general practitioner (GP) in primary care [13]. Work examining indications recorded in primary care around the time of the first antidepressant prescription and which secondary care specialists were seen has been done separately [14].

Antidepressant prescriptions for young people were increasing until a drop in 2002, and then began significantly increasing again from 2005 both in the UK [15–19] and other countries [20–23]. Increased prescribing of antidepressants could indicate a greater awareness and recognition of mental health and related issues and a willingness to seek help in the form of being diagnosed and/or treated. There is some evidence that in people of all ages, antidepressant prescribing varies by region [24] and between different ethnic groups in England [25,26]. Previous studies showing that CYP living in more deprived areas are more likely to receive antidepressant prescriptions have not taken region or ethnicity into account [15,16]. There is generally a lack of evidence about variation in antidepressant prescribing in CYP, particularly taking other possibly confounding factors into account. Differences in prescribing by deprivation, geographical location, and ethnicity could indicate differences in the distribution of risk factors for mental health disorders and/or differences in access to psychological therapies, which could both affect the likelihood of antidepressant prescribing.

It remains unclear, based on previous studies, whether the reported rise in antidepressant prescribing in CYP that began in 2005 is continuing and whether there is variation in prescribing across different groups. Our study aimed to examine changes over time and the variation in the use of antidepressant medicines in CYP aged between 5 and 17 years old between 1998 and 2017 in England. Our objectives were to 1) describe differences in antidepressant prescriptions for CYP over time by age, sex, deprivation, region, and ethnicity; 2) estimate variation in prescribing between these groups adjusting for the other factors; and 3) assess to what extent NICE guidelines for CYP are being adhered to.

## Methods

The full protocol for the study has previously been published [27]. Any analyses that were not prespecified in the protocol are described as sensitivity or post hoc analyses below. This study is reported as per the Strengthening the Reporting of Observational Studies in Epidemiology (STROBE) guideline (see S1 STROBE Checklist).

### Data sources

The cohort was extracted from a large primary care database (QResearch, version 43) linked to hospital episode statistics (HES) admitted patient care and outpatient data. At the time of the study, the QResearch database included health records of over 32 million patients from more than 1,500 general practices across the UK that record data using the Egton Medical Information Systems (EMIS) medical records computer system.

### Study participants

The study's open cohort was defined as all people registered on the QResearch database in England who were aged between 5 and 17 years between 1 January 1998 and 31 December 2017. Each person's study entry date was defined as the latest date of the following: 12 months after their registration with a study practice, 12 months after the installation date of their practice's EMIS computer system, 1 January of the year they turned 5 years old, or 1 January 1998. People were then followed up until the earliest date of them leaving the practice, dying, 1

January of the year they turned 18 years old, or the end of the follow-up period (31 December 2017).

## Outcomes

We extracted information on prescriptions for any antidepressant for each person in the cohort. In some cases, we had prescribing information from before a patient registered with a practice because of electronic transferring of prescription data or before the practice installed EMIS. In this way, we could identify prescriptions that took place before the study period. We examined all antidepressants combined and 4 different drug classes: selective serotonin reuptake inhibitors (SSRIs), tricyclic and related antidepressants (TCAs), serotonin and norepinephrine reuptake inhibitors (SNRIs), and other antidepressants, including monoamine oxidase inhibitors (MAOIs). Antidepressants included in each drug class can be found in S1 Table. We also considered individual antidepressant drugs separately. For the figures, the 10 most prescribed antidepressants overall (each representing at least 0.8% of prescriptions) were included. Any antidepressant prescribed within the study period was included in the prevalence analyses. Those with a record of an antidepressant prescription before their study entry were excluded for the incidence cohort so that only the first antidepressant prescription was examined.

## Covariates

We analysed 4 groups, defined by sex (female and male) and age (5–11 and 12–17 years). These age groups are similar to those specified in the NICE guidelines on depression in CYP [5] but exclude 18-year-olds, who may have been treated as adults. We also studied trends and variation for deprivation, different regions of England, and ethnic groups. Deprivation was measured using the Townsend deprivation index, an area-based measure of deprivation that combines information on 4 indicators (unemployment, non-car ownership, non-home ownership, household overcrowding) from the census [28]. Areas are then divided into quintiles based on their score. When ethnicity information was missing in QResearch, we supplemented this with the most recent valid ethnic code available in HES. We examined 5 broad ethnic groups: White, Mixed, Asian, Black, and Chinese or other ethnic group, plus those with no recorded ethnicity.

## Statistical analysis

We calculated incidence rates for being first prescribed an antidepressant and prevalence rates for people with a first or subsequent antidepressant prescription per 1,000 person-years for each year between 1998 and 2017. These were produced for the different sex and age groups for all antidepressants, the drug classes, and individual drugs as described above.

Incidence rate ratios were calculated using multilevel mixed-effects Poisson regression to take account of any clustering within GP practices for the different sex and age groups. The fully adjusted models included year, region, Townsend deprivation quintile, and ethnic group. In order to take account of varying patterns in antidepressant prescribing over time, linear trends were assessed for different periods using piecewise linear regression [29] with change points identified by previous studies: 2002 and 2005 [15,17]. The year 2008 was also identified as a change point for antidepressant prevalence in people aged 14 years and over, but not incidence [17]. We included this and assessed its statistical significance because our study has a longer follow-up than the previous analysis. Sensitivity analyses recoding the not known ethnic group to White and examining prescribing by deprivation after excluding London data were also performed after examining the initial results.

As a post hoc analysis, we assessed whether the incidence rate ratios (IRRs) for 12- to 17-year-olds from the Poisson regression were linked to other factors in each region. For this, we plotted the IRRs alongside previously published data from other sources: local authority spending on 'low-level' mental health services (those that are nonspecialist, preventive, and early intervention, which fall below specialist referral thresholds) per child in the 2018–2019 financial year in regions in England [30] and prevalence estimates of any depressive disorder and any anxiety disorder for each region from a national survey in England in 2017 of CYP aged 5–19 years [6].

All analyses were performed using Stata/SE v15 (StataCorp LLC, TX, USA).

### Ethics statement

The project was independently peer reviewed and accepted by the QResearch Scientific board and approved in accordance with the procedure agreed with the Trent Research Ethics Committee (reference: 18/EM/0400).

## Results

The flow chart detailing the selection of the cohort is shown in Fig 1. There were 4,349,638 CYP included in the prevalence study cohort. Of these, 14,537 (0.3%) had their first antidepressant prescription before their study entry date and were excluded from the incidence

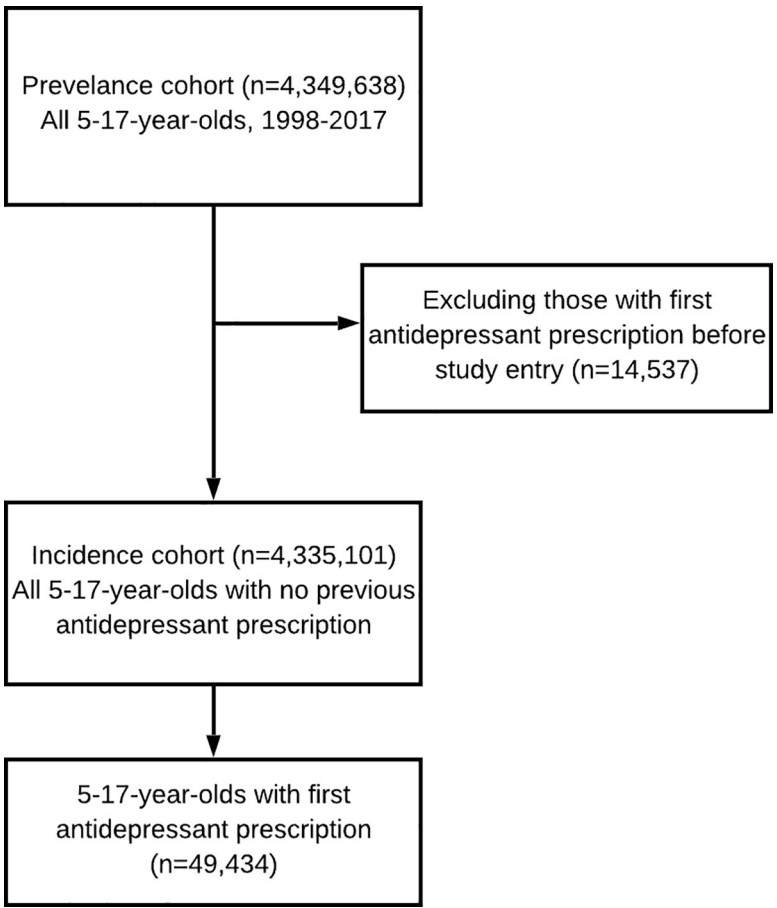

**Fig 1. Flowchart of selection of study participants.**

**Table 1. Characteristics of incidence cohort (incidence cohort excludes 14,537 patients with first antidepressant prescription before study entry) and the subset with a new prescription of any antidepressant during the study period, age 5 to 17 years, England 1998–2017.**

| | | Incidence Cohort | | Any New Antidepressant Prescribed Subcohort | |
|---|---|---|---|---|---|
| | | n | % | n | % |
| Total | | 4,335,101 | 100% | 49,434 | 100% |
| Age* | 5–11 years | 11.4 m person-years | | 5,133 | 10.4% |
| | 12–17 years | 9.0 m person-years | | 44,301 | 89.6% |
| Sex | Female | 2,080,843 | 48.0% | 32,571 | 65.9% |
| | Male | 2,254,258 | 52.0% | 16,863 | 34.1% |
| Townsend deprivation quintile | 1 (least deprived) | 973,529 | 22.5% | 11,943 | 24.2% |
| | 2 | 923,863 | 21.3% | 12,066 | 24.4% |
| | 3 | 863,260 | 19.9% | 10,856 | 22.0% |
| | 4 | 800,643 | 18.5% | 8,778 | 17.8% |
| | 5 (most deprived) | 761,722 | 17.6% | 5,687 | 11.5% |
| | Not known | 12,079 | 0.3% | 104 | 0.2% |
| Region | East Midlands | 206,792 | 4.8% | 2,816 | 5.7% |
| | East of England | 252,108 | 5.8% | 3,807 | 7.7% |
| | London | 1,056,707 | 24.4% | 5,723 | 11.6% |
| | North East | 148,885 | 3.4% | 1,943 | 3.9% |
| | North West | 672,537 | 15.5% | 7,973 | 16.1% |
| | South East | 887,291 | 20.5% | 13,261 | 26.8% |
| | South West | 430,169 | 9.9% | 5,866 | 11.9% |
| | West Midlands | 474,983 | 11.0% | 5,464 | 11.1% |
| | Yorkshire & Humber | 205,629 | 4.7% | 2,581 | 5.2% |
| Ethnicity | Not known | 1,710,263 | 39.5% | 15,853 | 32.1% |
| | Known | 2,624,838 | 60.5% | 33,581 | 67.9% |
| Ethnic group (% of known) | White | 1,975,726 | 75.3% | 30,731 | 91.5% |
| | Mixed | 86,678 | 3.3% | 566 | 1.7% |
| | Asian | 302,499 | 11.5% | 1,419 | 4.2% |
| | Black | 185,606 | 7.1% | 562 | 1.7% |
| | Chinese/Other | 74,329 | 2.8% | 303 | 0.9% |

*Number of person-years included given for incidence cohort by age, as some people were included in both age groups over the study period

cohort. This left 49,434 CYP who were first prescribed an antidepressant during the study period. Details of characteristics of the incidence cohort and the subcohort prescribed their first antidepressant are shown in Table 1. Almost three-quarters of the incidence cohort (nearly 3.2 million participants) entered the study aged between 5 to 11 years, and the median follow-up time was 3.6 years (interquartile range: 1.5–7.2 years). Ethnicity information was available in QResearch for 2,317,010 (53.5%) of the incidence cohort and was supplemented by HES records for a further 307,810 (7.1%).

Overall, SSRIs (62.6%) were the most commonly prescribed first antidepressant, followed by TCAs (35.7%) (Table 2). SNRIs (0.5%) and other antidepressants (1.3%) were rarely prescribed as the first antidepressant. The 5 most commonly prescribed first antidepressants were all either licensed for use in CYP (fluoxetine, imipramine, and sertraline) or mentioned in NICE guidelines (amitriptyline and citalopram, although citalopram is only recommended as

**Table 2. Number and percentage of CYP with first prescriptions in each drug class and for individual drugs during the study period, England 1998–2017, by age and sex.**

| | Total | | 5–11 Years | | | | 12–17 Years | | | |
|---|---|---|---|---|---|---|---|---|---|---|
| | | | Females | | Males | | Females | | Males | |
| | n | % | n | % | n | % | n | % | n | % |
| Any antidepressant | 49,434 | 100% | 1,845 | 100% | 3,288 | 100% | 30,726 | 100% | 13,575 | 100% |
| Drug class | | | | | | | | | | |
| SSRI | 30,949 | 62.6% | 298 | 16.2% | 594 | 18.1% | 20,895 | 68.0% | 9,162 | 67.5% |
| TCA | 17,624 | 35.7% | 1,538 | 83.4% | 2,674 | 81.3% | 9,285 | 30.2% | 4,127 | 30.4% |
| SNRI | 263 | 0.5% | 6 | 0.3% | 11 | 0.3% | 161 | 0.5% | 85 | 0.6% |
| Other | 640 | 1.3% | <5 | <0.3% | 9 | 0.3% | 414 | 1.3% | 214 | 1.6% |
| Individual drug | | | | | | | | | | |
| Fluoxetine | 17,493 | 35.4% | 171 | 9.3% | 335 | 10.2% | 11,940 | 38.9% | 5,047 | 37.2% |
| Amitriptyline | 12,181 | 24.6% | 709 | 38.4% | 921 | 28.0% | 7,627 | 24.8% | 2,924 | 21.5% |
| Citalopram | 5,828 | 11.8% | 29 | 1.6% | 43 | 1.3% | 4,244 | 13.8% | 1,512 | 11.1% |
| Sertraline | 5,612 | 11.4% | 83 | 4.5% | 164 | 5.0% | 3,404 | 11.1% | 1,961 | 14.4% |
| Imipramine | 3,323 | 6.7% | 718 | 38.9% | 1,564 | 47.6% | 424 | 1.4% | 617 | 4.5% |
| Paroxetine | 1,535 | 3.1% | 12 | 0.7% | 45 | 1.4% | 985 | 3.2% | 493 | 3.6% |
| Dosulepin | 734 | 1.5% | 5 | 0.3% | 6 | 0.2% | 490 | 1.6% | 233 | 1.7% |
| Escitalopram | 472 | 1.0% | <5 | <0.3% | <5 | <0.3% | 336 | 1.1% | 133 | 1.0% |
| Nortriptyline | 420 | 0.8% | 19 | 1.0% | 20 | 0.6% | 290 | 0.9% | 91 | 0.7% |
| Mirtazapine | 404 | 0.8% | <5 | <0.3% | 6 | 0.2% | 245 | 0.8% | 151 | 1.1% |
| Lofepramine | 289 | 0.6% | <5 | <0.3% | 7 | 0.2% | 216 | 0.7% | 62 | 0.5% |
| Venlafaxine | 225 | 0.5% | <5 | <0.3% | 10 | 0.3% | 133 | 0.4% | 78 | 0.6% |
| Trazodone | 165 | 0.3% | <5 | <0.3% | 23 | 0.7% | 79 | 0.3% | 60 | 0.4% |
| Fluvoxamine | 61 | 0.1% | <5 | <0.3% | <5 | <0.3% | 23 | 0.1% | 31 | 0.2% |
| Duloxetine | 38 | 0.1% | <5 | <0.3% | <5 | <0.3% | 28 | 0.1% | 7 | 0.1% |

**Abbreviations:** CYP, children and young people; SNRI, serotonin and norepinephrine reuptake inhibitor; SSRI, selective serotonin reuptake inhibitor; TCA, tricyclic and related antidepressant.

a second-line antidepressant in the treatment of depression). Over a third of first prescriptions were fluoxetine, a quarter were amitriptyline, and there were similar proportions of citalopram (11.8%) and sertraline (11.4%) prescriptions. Despite being licensed for treating obsessive-compulsive disorder in CYP, fluvoxamine was rarely prescribed as a first antidepressant in the study period, making up only 0.1% of new prescriptions.

TCAs accounted for over 80% of first antidepressant prescriptions in 5- to 11-year-olds, and imipramine and amitriptyline were the most commonly first-prescribed individual drugs in this age group. In adolescents, two-thirds of the newly prescribed antidepressants were SSRIs. Over the whole study period, fluoxetine (38.3%) and amitriptyline (23.8%) were the most commonly first-prescribed individual drugs in 12- to 17-year-olds.

## Incidence rates

Incidence rates for antidepressant prescribing showed distinct patterns in the age groups and were highest in 12- to 17-year-old females (S1 Fig).

Antidepressant prescriptions decreased over the study period in 5- to 11-year-olds. For TCAs, incidence rates decreased by 92% from a peak of 1.6 per 1,000 person-years in males and by 86% from 0.9 in females in 1999 to less than 0.14 per 1,000 for both sexes in 2017 (Fig 2). SSRI incidence rates, however, increased from 0.05 per 1,000 person-years in females and

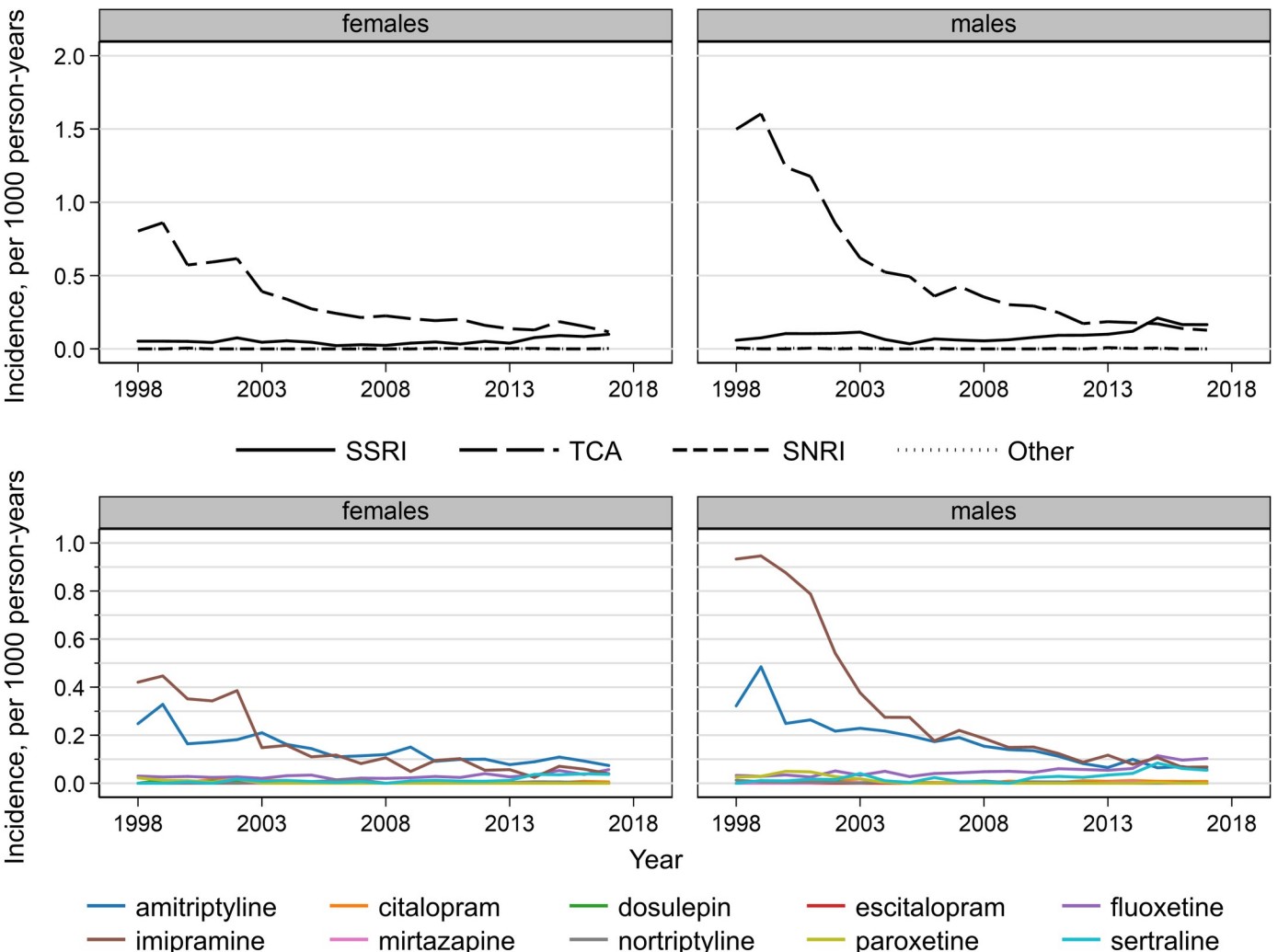

**Fig 2. Antidepressant drug class and individual drug incidence rates per 1,000 person-years in 5- to 11-year-olds, England, 1998–2017, by sex.** SNRI, serotonin and norepinephrine reuptake inhibitor; SSRI, selective serotonin reuptake inhibitor; TCA, tricyclic and related antidepressant

0.06 in males in 1998 to 0.10 in females and 0.17 in males in 2017. Imipramine was initially the most commonly prescribed individual drug for 5- to 11-year-olds. These rates decreased and were similar to amitriptyline from 2003 in females and 2006 in males. Fluoxetine and sertraline incidence rates increased over the study period so that these 4 drugs had similar incidence rates in 2017 (0.04–0.07 per 1,000 person-years in females and 0.05–0.10 in males).

For 12- to 17-year-olds, antidepressant incidence rates were 2.2 and 2.7 times higher in 2017 than 2005 in females and males, respectively (S1 Fig). TCA incidence rates declined over the study period to 2.0 per 1,000 person-years in females and 0.7 in males in 2017 (Fig 3). SSRI incidence rates increased between 1998 and 2002, decreased until 2005, and then increased again after this reaching a rate of 3.5 per 1,000 in males and 7.6 per 1,000 in females in 2017. The SSRI incidence rates per 1,000 person-years in 1998, 2002, and 2005 were 3.7, 7.1, and 2.5 in females and 1.1, 2.1, and 0.8 in males. Fluoxetine, sertraline, amitriptyline, and citalopram have been the 4 most commonly prescribed first antidepressants in 12- to 17-year-olds since 2003 in females and 2008 in males. This is due to a sharp decrease in paroxetine prescribing after 2002 and decreasing imipramine prescribing. The largest absolute increases were in

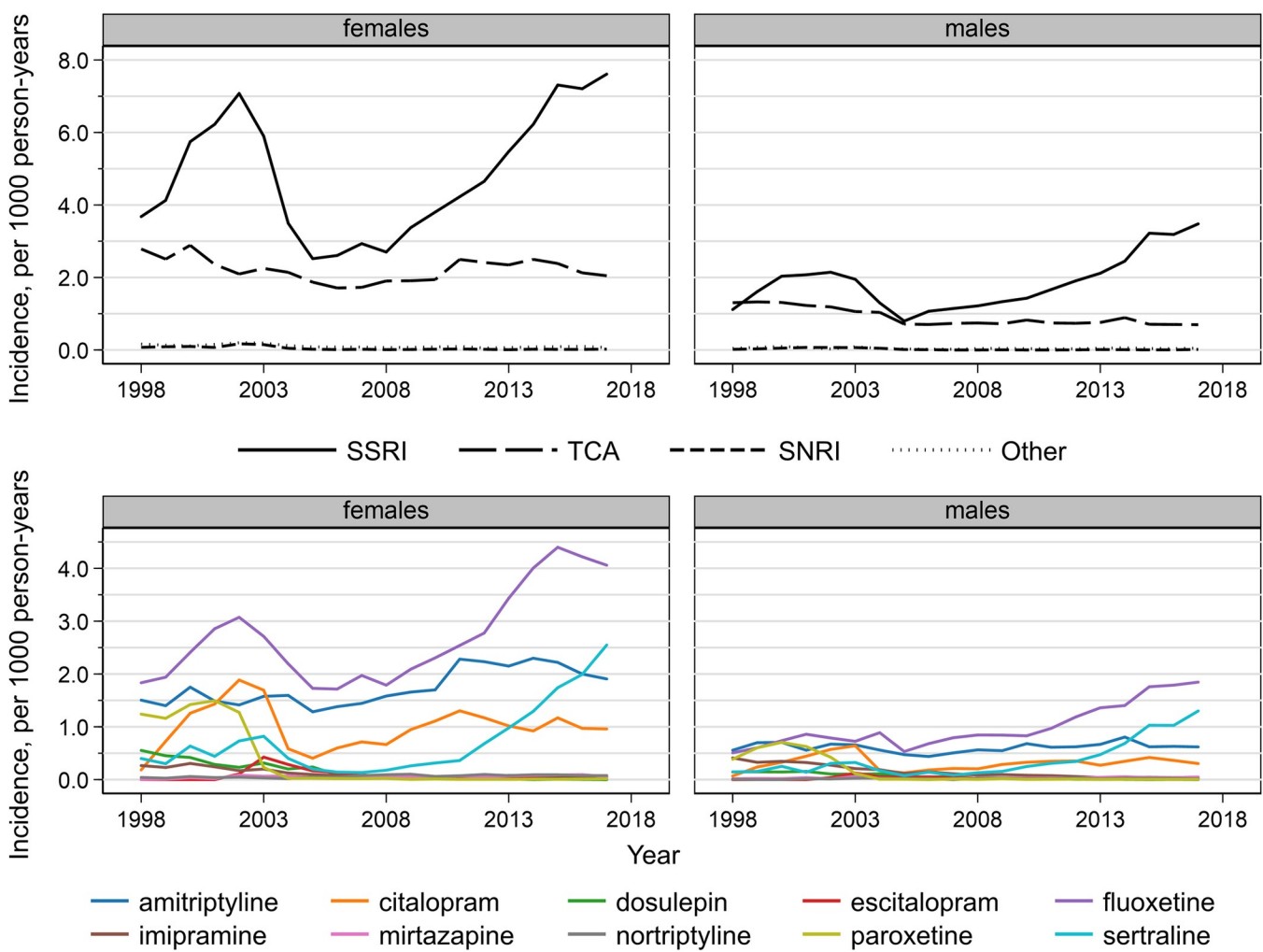

**Fig 3. Antidepressant drug class and individual drug incidence rates per 1,000 person-years in 12- to 17-year-olds, England 1998–2017, by sex.** SNRI, serotonin and norepinephrine reuptake inhibitor; SSRI, selective serotonin reuptake inhibitor; TCA, tricyclic and related antidepressant

fluoxetine and sertraline prescriptions in males and females, with sertraline becoming the second most commonly first-prescribed antidepressant for males in 2015 and for females in 2017.

S2–S4 Figs show the crude antidepressant incidence rates over time for 12- to 17-year-olds by deprivation, region, and ethnicity. These rates were lower for those in the most deprived group (S2 Fig) and in London than in other areas (S3 Fig). White males and females had the highest crude incidence rates and Black and Asian groups the lowest (S4 Fig).

## IRRs

Fully adjusted IRRs for antidepressant prescribing are shown in Table 3. In 5- to 11-year-olds, the incidence decreased throughout the time period, with the largest decrease per year between 2002 and 2005. For 12- to 17-year-olds, there was an increase in incidence rates per year from 1998 until 2002, then a decrease until 2005, a small increase until 2008, and then another increasing trend per year similar in magnitude to the first period.

**Table 3. IRRs for any antidepressant by age and sex fully adjusted for all variables shown and accounting for clustering by GP practice, England 1998–2017.**

| | | 5–11 Years | | | | | | 12–17 Years | | | | | |
|---|---|---|---|---|---|---|---|---|---|---|---|---|---|
| | | Females | | | Males | | | Females | | | Males | | |
| | | IRR | 95% CI | p | IRR | 95% CI | p | IRR | 95% CI | p | IRR | 95% CI | p |
| Trends within periods (per year) | 1998–2002 | 0.92 | (0.87–0.97) | 0.001 | 0.87 | (0.84–0.90) | <0.001 | 1.10 | (1.08–1.12) | <0.001 | 1.08 | (1.05–1.10) | <0.001 |
| | 2002–2005 | 0.78 | (0.73–0.84) | <0.001 | 0.78 | (0.74–0.82) | <0.001 | 0.76 | (0.74–0.77) | <0.001 | 0.77 | (0.74–0.79) | <0.001 |
| | 2005–2008 | 0.92 | (0.85–0.98) | 0.015 | 0.91 | (0.86–0.96) | <0.001 | 1.03 | (1.01–1.05) | 0.002 | 1.01 | (0.98–1.04) | 0.397 |
| | 2008–2017 | 0.97 | (0.95–1.00) | 0.036 | 0.95 | (0.94–0.97) | <0.001 | 1.09 | (1.08–1.09) | <0.001 | 1.08 | (1.08–1.09) | <0.001 |
| Townsend deprivation quintile | Q1 (least deprived) | 1.00 | | | 1.00 | | | 1.00 | | | 1.00 | | |
| | Q2 | 1.09 | (0.95–1.24) | 0.226 | 0.96 | (0.87–1.06) | 0.432 | 1.15 | (1.11–1.18) | <0.001 | 1.14 | (1.09–1.20) | <0.001 |
| | Q3 | 1.09 | (0.95–1.26) | 0.223 | 1.11 | (1.00–1.23) | 0.053 | 1.30 | (1.26–1.35) | <0.001 | 1.23 | (1.17–1.30) | <0.001 |
| | Q4 | 1.25 | (1.08–1.46) | 0.004 | 1.18 | (1.05–1.32) | 0.005 | 1.46 | (1.40–1.52) | <0.001 | 1.37 | (1.29–1.45) | <0.001 |
| | Q5 (most deprived) | 1.33 | (1.11–1.59) | 0.002 | 1.13 | (0.98–1.30) | 0.089 | 1.48 | (1.41–1.55) | <0.001 | 1.29 | (1.20–1.39) | <0.001 |
| | Not known | 1.51 | (0.67–3.39) | 0.315 | 1.27 | (0.66–2.45) | 0.480 | 1.12 | (0.88–1.44) | 0.349 | 1.18 | (0.79–1.76) | 0.428 |
| | Deprivation trend (excluding not known) | 1.07 | (1.03–1.12) | 0.001 | 1.05 | (1.02–1.08) | 0.002 | 1.12 | (1.11–1.13) | <0.001 | 1.08 | (1.07–1.10) | <0.001 |
| Region | East Midlands | 1.00 | | | 1.00 | | | 1.00 | | | 1.00 | | |
| | East of England | 0.71 | (0.49–1.04) | 0.077 | 0.77 | (0.56–1.07) | 0.116 | 1.18 | (1.02–1.36) | 0.027 | 1.12 | (0.95–1.32) | 0.191 |
| | London | 0.50 | (0.37–0.69) | <0.001 | 0.48 | (0.37–0.63) | <0.001 | 0.44 | (0.39–0.49) | <0.001 | 0.52 | (0.45–0.60) | <0.001 |
| | North East | 0.93 | (0.61–1.40) | 0.716 | 0.86 | (0.60–1.24) | 0.422 | 0.76 | (0.64–0.89) | 0.001 | 0.75 | (0.62–0.91) | 0.004 |
| | North West | 0.74 | (0.54–1.01) | 0.062 | 0.66 | (0.50–0.87) | 0.003 | 0.81 | (0.72–0.91) | 0.001 | 0.90 | (0.78–1.04) | 0.153 |
| | South East | 1.03 | (0.76–1.39) | 0.861 | 1.08 | (0.83–1.40) | 0.570 | 1.20 | (1.06–1.35) | 0.003 | 1.29 | (1.12–1.48) | <0.001 |
| | South West | 0.87 | (0.63–1.22) | 0.422 | 0.83 | (0.62–1.11) | 0.205 | 1.10 | (0.96–1.25) | 0.162 | 1.02 | (0.87–1.19) | 0.827 |
| | West Midlands | 0.80 | (0.57–1.11) | 0.174 | 0.82 | (0.62–1.09) | 0.173 | 0.86 | (0.76–0.98) | 0.024 | 0.94 | (0.81–1.09) | 0.433 |
| | Yorkshire & Humber | 0.99 | (0.68–1.45) | 0.971 | 0.98 | (0.71–1.36) | 0.913 | 0.81 | (0.69–0.94) | 0.006 | 0.77 | (0.65–0.93) | 0.006 |
| Ethnic group | White | 1.00 | | | 1.00 | | | 1.00 | | | 1.00 | | |
| | Mixed | 0.75 | (0.50–1.15) | 0.187 | 0.85 | (0.62–1.17) | 0.319 | 0.66 | (0.59–0.74) | <0.001 | 0.81 | (0.70–0.94) | 0.007 |
| | Asian | 0.70 | (0.54–0.89) | 0.004 | 0.64 | (0.52–0.78) | <0.001 | 0.41 | (0.38–0.44) | <0.001 | 0.47 | (0.42–0.52) | <0.001 |
| | Black | 0.71 | (0.51–0.98) | 0.038 | 0.56 | (0.42–0.75) | <0.001 | 0.32 | (0.29–0.36) | <0.001 | 0.32 | (0.27–0.38) | <0.001 |
| | Chinese/Other | 0.49 | (0.26–0.91) | 0.025 | 0.47 | (0.28–0.79) | 0.004 | 0.44 | (0.38–0.51) | <0.001 | 0.57 | (0.47–0.69) | <0.001 |
| | Not known | 0.52 | (0.47–0.58) | <0.001 | 0.54 | (0.49–0.58) | <0.001 | 0.45 | (0.44–0.46) | <0.001 | 0.41 | (0.39–0.42) | <0.001 |

**Abbreviations:** CI, confidence interval; GP, general practitioner; IRR, incidence rate ratio.

Antidepressant incidence rates increased with increasing deprivation in all 4 age–sex subgroups after adjustment. The strongest trend was seen in 12- to 17-year-old females (adjusted IRR = 1.12, 95% confidence interval [CI] 1.11–1.13, p < 0.001, per deprivation quintile increase). However, unadjusted estimates of antidepressant prescribing by deprivation showed a statistically significant trend of decreasing prescribing with increasing deprivation, driven by the most deprived quintile, in all age and sex groups apart from 5- to 11-year-old females (S3 Table). For the older age group, adjusting for region resulted in increased prescribing rates with increasing deprivation, whereas adjusting for year alone did not affect the unadjusted trend results, and adjusting for just ethnicity attenuated the association but showed those living in more deprived areas were still less likely to be prescribed antidepressants. This statistically significant pattern with deprivation was also shown for 5- to 11-year-old females after adjusting for region. This association was only evident after taking GP practice clustering into account in the fully adjusted models (Table 3) in 5- to 11-year-old males. Within London, there was a statistically significant unadjusted association with deprivation and prescribing in 12- to 17-year-olds, with those living in more deprived areas having lower prescribing rates.

After excluding London in the unadjusted analyses, increased prescribing with increasing deprivation was found for all groups apart from 5- to 11-year-old males, highlighting the influence of region on these results.

London had the lowest IRRs and the South East had the highest IRRs in all sex and age subgroups. The South East estimates were all more than double that of London. The North East had the second lowest prescribing rate estimates for 12- to 17-year-olds. These estimates were 1.4 (in males) and 1.7 (in females) times higher than the London estimates.

Asian, Black, and Chinese or other ethnic groups, as well as those whose ethnicity was not known, had statistically significantly lower prescription incidence rates compared with the White baseline group in 5- to 11-year-olds. For the older age group, the Mixed ethnic group also had significantly lower incidence rates than the White group. In both males and females, the lowest IRRs in adolescents were found in the Black group (IRR 0.32, 95% CI 0.27–0.38, $p < 0.001$ and IRR 0.32, 95% CI 0.29–0.36, $p < 0.001$, respectively) compared with the White group. Results were attenuated in the sensitivity analysis in which the not known ethnic group were recoded as White (S2 Table). There were no statistically significant differences between all other ethnic groups and the White/not known baseline for females aged 5–11 years. In males aged 5–11 years in the Chinese or other (IRR 0.59, 95% CI 0.36–0.99, $p = 0.046$), Black (IRR 0.72, 95% CI 0.53–0.96, $p = 0.027$), and Asian (IRR 0.83, 95% CI 0.68–1.01, $p = 0.067$) ethnic groups, the IRRs were still lower than the White/not known group, although the Asian group was only borderline statistical significance. In the older age group, the attenuated results showed the same patterns, with only the male Mixed ethnic group no longer statistically significantly different from the baseline.

## Prevalence rates

The prevalence rate patterns overall, in antidepressant drug classes, and in individual drugs were similar to the incidence rates (S1, S5 and S6 Figs). Prevalence estimates for any antidepressant prescriptions were highest for 5- to 11-year-olds in 1999 and decreased over the study period. Despite a peak in the early 2000s in prescriptions for 12- to 17-year-olds, the highest rates for males and females were in 2017 (7.9 and 16.4 per 1,000 person-years, respectively). In 12- to 17-year-olds, there were an extra 6.6 females and 3.7 males per 1,000 person-years who had a second or later antidepressant prescription in 2017. There were very small differences between the incidence and prevalence rates in 5- to 11-year-olds throughout the study period.

S7 Fig shows a negative association between spending per child on 'low-level' mental health services and the prescribing patterns from our study. There appears to be no strong correlation between either spending or prescribing and the depression and anxiety prevalence estimates.

## Discussion

### Summary of main results

This study has shown a diverging pattern of decreasing prescribing of antidepressants in 5- to 11-year-olds between 1998 and 2017, whilst rates have more than doubled in 12- to 17-year-olds since 2005. The most commonly first-prescribed antidepressants are either licensed for use in CYP or included in national guidelines. Rates of antidepressant prescribing were higher for CYP living in more deprived areas after accounting for GP practice clustering and region. Children and adolescents living in South East England were more likely to be prescribed antidepressants, and those living in London the least likely. Prescribing rates were highest in White and lowest in Black adolescents.

## Comparison with other studies

This study has shown that the increase in antidepressant prescribing in CYP in England from 2005 previously found [15–19] has continued until 2017. The 2017 antidepressant prescribing prevalence rates in adolescents from our study (0.79% and 1.64%) are around half the associated depressive disorder estimates (1.6% and 3.8%) [6] in males and females, respectively. Citalopram prescribing has previously been found to be higher than in our study—higher than fluoxetine in 6- to 18-year-olds [16], similar to fluoxetine in under-18s in 2009 [15], and the second most commonly prescribed antidepressant after fluoxetine in people aged under 20 [20]. Whilst the inclusion of those aged 18 years [16] and over [20] who may have been treated as adults may explain the higher rates, other methodological differences, such as variation in factors not accounted for in the unadjusted analyses, could also be important.

A previous study reported that for people of all ages in England, the prevalence of antidepressant prescriptions was lowest in London and highest in the North East of England [24]. We also found the lowest incident prescription rates in London; however, the North East was the next lowest area for adolescents. There could be differences in antidepressant prescribing patterns between CYP and adults, including initiating prescribing and the number of prescriptions because of the length of time people stay on antidepressants affecting the prevalence analysis, that account for this difference. Our study examined prescriptions, whilst Grigoroglou and colleagues [24] used dispensing data, so differences in whether patients filled their prescriptions may also exist.

A report by the Children's Commissioner into early access to mental health support found variation in the stated spending on 'low-level' preventive and early-intervention (including nonspecialist psychological support services) services between different regions [30]. London had the highest spending, and the East of England spent the least per child. We found that the lowest prescribing for adolescents was in London, which is located between the South East and East of England regions. These 2 areas had the highest prescribing IRR estimates for 12- to 17-year-olds. S7 Fig shows there appears to be a negative association between spending per child on 'low-level' mental health services (London highest; East of England and South East lowest) and the prescribing patterns from our study (London lowest; East of England and South East highest). This implies that antidepressants might be prescribed more frequently to CYP in areas where there is less investment in preventive services or psychological support available, although further research is needed to test this hypothesis. That there was no strong correlation between either spending or prescribing and the depression and anxiety prevalence estimates could be due to differences in age groups and years studied.

Our study has good face validity because we found similar trends to those published elsewhere. For example, we found that after adjustment for GP practice clustering and region, CYP living in more deprived areas are more likely to receive antidepressant prescriptions compared with those in the less deprived areas, which has previously been shown for children and adolescents [15,16] and adults [24] in the UK. Although the unadjusted results showed lower antidepressant prescribing in adolescents living in the most deprived quintile (S2 Fig), this was due to the lower prescribing and large proportion of people living in the most deprived quintile in London. Excluding London from an unadjusted analysis of trend over deprivation changed the estimates in 12- to 17-year-olds from 0.96 (p < 0.001) in both sexes to 1.06 in females and 1.05 in males (both p < 0.001) (S3 Table). Measures of deprivation were not found to be statistically significantly associated with spending on CYP mental health per child at the Clinical Commissioning Group level in a study by Rocks and colleagues [31]. This study included early help and targeted services (tier 2) and specialised Child and Adolescent Mental Health Services (CAMHS) (tier 3). Spending on early interventions by nonspecialists (tier 1) could influence antidepressant prescriptions and might be linked to deprivation.

## Strengths and limitations

The main strengths of this study are its size, representativeness, and duration. We analysed data from over 4.3 million CYP across England over a 20-year period, which, to our knowledge, is the largest study to date on antidepressant prescribing in CYP. The vast majority of the UK population are registered with a GP practice [13], and the QResearch database is the most nationally representative primary care database in England [32] with comprehensive information on prescriptions. This means that the results are likely to generalise well within the UK. We have been able to examine a long time series for different sex and age groups and investigate variation by deprivation, region, and ethnicity whilst taking all these factors into account.

A limitation of the study is that it does not include any secondary care prescriptions. There is no publicly available information about antidepressant prescribing for CYP in secondary care in the UK. Most parents would initially first visit a GP with concerns about their child's mental health issues [33]. The GP would refer them to a specialist in child and mental health following if appropriate [5,10]. Furthermore, in the UK, GPs are typically responsible for the ongoing prescribing of antidepressant medicines if initiated by a specialist. It is possible that what we have identified as a first prescription is actually a subsequent prescription in primary care following an initial prescription in secondary care. Without secondary care prescriptions, the prevalence estimates will be underestimates of the total antidepressant prescriptions in CYP. However, the prevalence patterns should be unaffected by who initiated the first prescription if prescribing is transferred to primary care. We were only able to assess whether antidepressants were prescribed, not whether they were dispensed or taken. Unmeasured confounding may also still be present. We chose to include all CYP regardless of whether they had an appropriate indication recorded and cannot be sure that particular antidepressants were prescribed for the indications they are licensed for.

Because we used data routinely recorded in primary care rather than prospectively collecting the data specifically for the study, not all information we required was available. Despite using all available ethnicity information from QResearch and HES records, ethnicity was still missing for 40%. As a sensitivity analysis, we recoded those with no ethnicity information to a new White/not known group. This attenuated the results for the other ethnic groups, but these were still statistically significant for females aged 12–17 years, males aged 12–17 years (apart from the Mixed group), and Black and Chinese or other ethnic groups in males aged 5–11 years.

## Implications

Our study was not designed to determine whether increased prescribing of antidepressants in CYP since 2005 is due to increasing rates of mental health problems, greater awareness and help-seeking for those with mental health issues, prescribing behaviour, patient choice, or because there are issues with accessing other psychological therapies [6]. Generally, the antidepressants prescribed to CYP appear to be those licensed for use in under-18s in the UK or listed for use with particular indications in the British National Formulary for Children [8] despite not being licensed. The rapid recent increase of sertraline as the first antidepressant prescribed is of interest. Sertraline is licensed in the UK for use in CYP for obsessive-compulsive disorder and recommended as a second-line treatment for major depression in CYP by NICE [5]. The prevalence of obsessive-compulsive disorder is low in England (0.4% in 2017) [6], and as a second-line drug, sertraline would not be expected to be the first antidepressant prescribed for major depression. There are currently no UK guidelines on treating CYP with anxiety, although meta-analyses have shown SSRIs, including sertraline, to be effective [34]. General practitioners may be following the adult guidance, which suggests sertraline and was published in 2011 [35], after which female adolescent sertraline prescription rates rapidly increased. The relatively high prescribing rates of amitriptyline should be explored further to

determine what indications this drug is being prescribed for. Whilst amitriptyline is unlicensed for use in CYP, it is included in NICE guidelines for treating headaches [12] and the British National Formulary for Children for treating neuropathic pain [8]. TCAs in general are not recommended for depressive disorder in CYP and those who self-harm [36] because of life-threatening cardiac risks associated with overdose, but rates of self-harm have been shown to be increasing in 13- to 16-year-olds in recent years in the UK [37]. Understanding which indications are recorded around the time of these first antidepressant prescriptions is important, and work examining this has been analysed separately [14].

Future work should further examine whether any barriers in awareness, access to appropriate healthcare services, and treatment exist, and if so, whether they vary for different groups and how they can be tackled. Determining whether increased spending on preventive, early-intervention services and access to psychological therapies leads to lower antidepressant prescription rates, and how these services compare in terms of outcomes is an important area of work that could lead to possible savings in healthcare spending and improved health for the population.

## Supporting information

**S1 STROBE Checklist. STROBE Statement for 'Incidence and prevalence of primary care antidepressant prescribing in children and young people in England, 1998–2017: a population-based cohort study'.**
(DOCX)

**S1 Table. Antidepressants included in each drug class.**
(XLSX)

**S2 Table. IRRs for any antidepressant for recoded ethnic groups fully adjusted for year, deprivation, and region and accounting for clustering by GP practice, England 1998–2017, by age and sex.** GP, general practitioner; IRR, incidence rate ratio
(XLSX)

**S3 Table. Unadjusted IRRs for any antidepressant for deprivation overall, London only, and excluding London and total population adjusted separately for region, year, and ethnicity, England 1998–2017, by age and sex.** IRR, incidence rate ratio
(XLSX)

**S1 Fig. Overall antidepressant drug incidence and prevalence rates per 1,000 person-years, England 1998–2017, by age and sex.**
(TIF)

**S2 Fig. Antidepressant group incidence rates and 95% CIs per 1,000 person-years in 12- to 17-year-olds, England 1998–2017, by Townsend deprivation quintile (excluding not known) and sex.** CI, confidence interval; Townsend Q1, least deprived quintile; Townsend Q5, most deprived quintile
(TIF)

**S3 Fig. Antidepressant group incidence rates and 95% CIs per 1,000 person-years in 12- to 17-year-olds, England 1998–2017, by region and sex.** CI, confidence interval
(TIF)

**S4 Fig. Antidepressant group incidence rates and 95% CIs per 1,000 person-years in 12- to 17-year-olds, England 1998–2017, by ethnic group and sex.** CI, confidence interval
(TIF)

**S5 Fig. Antidepressant drug class and individual drug prevalence rates per 1,000 person-years in 5- to 11-year-olds, England 1998–2017, by sex.**
(TIF)

**S6 Fig. Antidepressant drug class and individual drug prevalence rates per 1,000 person-years in 12- to 17-year-olds, England 1998–2017, by sex.**
(TIF)

**S7 Fig.  Regional (A) fully adjusted IRRs for any antidepressant, age 12–17 for males and females, 1998–2017; (B) LA spending on 'low-level' mental health services per child, 2018– 2019 [30]; (C) prevalence of any depressive disorder, age 5–19, 2017 [6]; and (D) prevalence of any anxiety disorder, age 5–19, 2017 [6].** IRR, incidence rate ratio; LA, Local Authority.
(TIF)

# Acknowledgments

The authors thank patients and EMIS practices who contribute to the QResearch database, EMIS and the University of Nottingham for expertise in establishing and developing the QResearch database, and the University of Oxford for its continued support and development. The HES data used in this analysis are reused by permission from National Health Service (NHS) Digital, who retain the copyright in that data. NHS Digital bear no responsibility for the analysis or interpretation of the data.

The views expressed are those of the authors and not necessarily those of the NHS, the National Institute for Health Research (NIHR), or the Department of Health and Social Care.

# Author Contributions

**Conceptualization:** Ruth H. Jack, Chris Hollis, Carol Coupland, Richard Morriss, Roger David Knaggs, Debbie Butler, Andrea Cipriani, Samuele Cortese, Julia Hippisley-Cox.

**Data curation:** Ruth H. Jack, Julia Hippisley-Cox.

**Formal analysis:** Ruth H. Jack.

**Funding acquisition:** Chris Hollis.

**Methodology:** Ruth H. Jack, Carol Coupland, Julia Hippisley-Cox.

**Supervision:** Chris Hollis, Carol Coupland, Julia Hippisley-Cox.

**Writing – original draft:** Ruth H. Jack.

**Writing – review & editing:** Ruth H. Jack, Chris Hollis, Carol Coupland, Richard Morriss, Roger David Knaggs, Debbie Butler, Andrea Cipriani, Samuele Cortese, Julia Hippisley-Cox.

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
