## [Editor Report · Decision Letter 0]

24 Feb 2020

Dear Dr Jack, 

Thank you for submitting your manuscript entitled "Trends and variation in the incidence and prevalence of antidepressant prescribing in children and young people in England: a population-based cohort study" for consideration by PLOS Medicine.

Your manuscript has now been evaluated by the PLOS Medicine editorial staff and I am writing to let you know that we would like to send your submission out for external peer review.

Kind regards,

Helen Howard, for Clare Stone PhD 

Acting Editor-in-Chief

PLOS Medicine 

plosmedicine.org

---

## [Decision Letter · Decision Letter 1]

6 Apr 2020

Dear Dr. Jack,

Thank you very much for submitting your manuscript "Trends and variation in the incidence and prevalence of antidepressant prescribing in children and young people in England: a population-based cohort study" (PMEDICINE-D-20-00542R1) for consideration at PLOS Medicine. 

Your paper was discussed among the editorial team and sent to independent reviewers, including a statistical reviewer. The reviews are appended at the bottom of this email and any accompanying reviewer attachments can be seen via the link below:

[LINK]

In light of these reviews, we will not be able to accept the manuscript for publication in the journal in its current form, but we would like to invite you to submit a revised version that fully addresses the reviewers' and editors' comments. You will appreciate that we cannot make a decision about publication until we have seen the revised manuscript and your response, and we expect to seek re-review by one or more of the reviewers. 

We hope to receive your revised manuscript by Apr 27 2020 11:59PM. Please email us (plosmedicine@plos.org) if you have any questions or concerns.

Please let me know if you have any questions. Otherwise, we look forward to receiving your revised manuscript shortly. 

Sincerely,

Richard Turner, PhD

rturner@plos.org

Noting PLOS' data policy, please adapt your data statement to briefly explain the criteria that will be applied to requests for access to study data, and include a non-author contact. 

Please begin the title with "Incidence and prevalence of ...", quote the study period, and add "in primary care" if appropriate. 

In your abstract, alongside the sizes of the cohorts studied, please quote aggregate participant characteristics from table 1. 

Please add some additional quantitative details on study findings to your abstract, for example to support statements such as "... antidepressant prescribing decreased in 5- to 11-year olds". 

To the "methods and findings" subsection of your abstract, please add a new final sentence quoting 2-3 of the study's main limitations. 

After the abstract, we will need to ask you to add a new and accessible "author summary" section in non-identical prose. You may find it helpful to consult one or two recent research papers published in PLOS Medicine to get a sense of the preferred style. 

At line 368, please broaden the discussion of study limitations to cite relevant weaknesses of observational studies, for example. 

Throughout the paper, please quote exact p values or p<0.001. Where available, please quote p values alongside 95% CI. 

Please remove trademarks throughout the article. 

Please add institutional author names to references where appropriate, e.g., reference 1. 

Please add accessed dates for online references. 

Please add a completed checklist for the most appropriate reporting guideline, which we imagine will be STROBE or RECORD, as a supplementary file (referred to in your methods section). In the checklist, please refer to individual items by section (e.g., "Methods") and paragraph number rather than by line or page numbers, as the latter generally change in the event of publication. 

Comments from the reviewers:

*** Reviewer #1: 

I found this to be an interesting and well written article. The topic is of considerable public health importance and of wider societal concern, for the UK and for many other countries. The data source utilised is robust and the study design and statistical analyses seem appropriate. I have just a few minor critical comments, which are listed below.

1) It would be preferable to include at least one implication (for clinicians, public health experts, policymakers or researchers) in the 'Conclusions' subsection of the abstract.

2) The authors ought to say more about the risks associated with prescribing of amitriptyline and other tricyclic antidepressants given that: a) the drug is highly toxic in overdose, and the National Institute for Health and Care Excellence (NICE) recommended in November 2011 that the drug should not be prescribed to anyone with a history of self-harm (NICE, Clinical Guideline no. 133); b) incidence of self-harm has risen sharply among 13-16 year olds in the UK from 2011 (Morgan et al. BMJ 2017;359:j4351). 

3) In the Covariates subsection of the Methods, the authors make brief mention of missing data as regards the ethnicity variable that was available in the QResearch dataset. Further on in the manuscript, in the 'Strengths and limitations' subsection of the Discussion, information is given concerning the amount of data that are missing for this variable. It would be ideal if this more detailed information was provided in the Methods. Likewise, for any tables that report information pertaining to ethnicity, it would be useful for readers to be informed about the level of missing data for this variable via footnotes.

4) Given that they are geographically adjacent, it is an intriguing finding that the lowest prescription incidence rate observed was for London and the highest was for the South East. Many readers will likely be unaware that these two regions both lie in the South East of England, and so I suggest that this is explained. 

*** Reviewer #2: 

This is a pharmacoepidemiological study, using a general practice database in the UK (QResearch) to examine prevalence, incidence and associated temporal trends in antidepressant prescribing for 5-17 year olds. 

A strength of the paper is its topical nature: data are available up to 2017. It has a large sample size and the sample is believed to be representative. While the analysis does not assess duration of therapy, it provides some relevant information on this topic since both incidence and prevalence estimates are presented. The trends over time are complex, and this is addressed with a piecewise regression approach in the multivariable component of the analysis. 

The study is highly descriptive. Information on the reasons for prescribing are apparently not available, so the results (as the authors point out) are subject to various interpretations (e.g. see the non-specific interpretive statement at the start of the "implications" section. Nevertheless, it is important to understand patterns of drug use, and associated trends, even if this mainly generates hypotheses for additional studies. In addition to describing trends, model-based estimates are presented, which produce some interesting results. 

The main findings are that there are differing trends in the 5-11 (decreasing) versus the 12-17 (increasing) age groups, and that the most commonly prescribed medications are those recommended by guidelines.

Another important finding is that the increase in antidepressants use in the 12-17 year olds has been substantial (a 2-3 fold increase between 2005 and 2017), which is important to know. Unfortunately, it is difficult to interpret since the absolute rates remain low, so the increase may either be a good or a bad thing. This is another example of where the study generates interesting hypotheses that should be address by additional research. Lower incidence in ethnic groups (Black and Asian) point towards possible issues of stigma and/or literacy in these groups.

Unadjusted estimates showed decreased prescribing in areas with increased deprivation whereas the opposite trend is seen in the adjusted analysis. Obviously some form of confounding (due to more ethnic minorities living in the most deprived areas?) is occurring but since the authors report only omnibus adjustments for multiple factors, it is not possible for readers to identify the source of this confounding. 

The authors make some interesting "ecological" comparisons, which leads to the generation of additional hypotheses. For, example, the possibility that low access to basic mental health care in regions that spend little on "low level" care, may lead to higher incidence of antidepressant use. 

The incidence data is described as "first" use of an antidepressant, but the data collection appears to go only 12 months prior to cohort entry for some patients, so the study appears not to confirm first use. 

In Table 3, the (trends within periods) rows doesn't seem to have a baseline group, so I'm not sure how to interpret the reported IRRs. 

Overall, I feel that this is a good descriptive study that provides important information. 

*** Reviewer #3: 

Comments on ms PMEDICINE-D-20-00542

"Trends and variations in the incidence and prevalence of antidepressant prescribing in children and young people in England: a population-based cohort study

This article provides information about antidepressant prescription in the age group 5-17 yrs in England. The data source is a primary care database containing anonymised healthcare records from primary care, with an incidence cohort of 4.3 million children and young people. 

Some comments:

The size of the database is impressive; however, I do not fully understand the degree to what this database covers the population of interest, i.e. children and adolescents <18 yrs of age with mental health problems/depression. The authors acknowledge that they do not have access to data from the specialist level mental health services, but could they describe how the health care system in the UK is organized? Where would a family turn to seek help for a depressed youth? How likely is it that your first visit would be in primary care? In many other countries, pharmacological treatment of depressed youth would be the sole responsibility of the specialist level services, and primary care would refer youth in need of pharmacological treatment to the specialist services. Is this different in the UK?

Without this information, it is impossible to assess the relevance of the results, both in regard to how generalizable they are to the entire UK population but also to other contexts outside the UK. A related question is if the geographical variation seen in the results could be explained by access to specialist level services (e.g. is that more accessible in the London area than in other parts of the UK)?

I ran across an article on the UK primary care database (Moore ea, BMJ 2009) that may be of relevance. 

The authors have added a reference in the introduction that shows that the waitlist to see a mental health specialist was long in 2017. How would a "long wait-list condition" affect patterns of seeking help? Would it be likely that youth instead contacted primary care, or would it lead to a delay in treatment?

In this regard, please explain what "low-level services" mean and, if possible, how the organisation of the service system may have influenced the results (or how we can understand them). Does a "low-level service" provide only psychosocial interventions, or do they also prescribe meds? Would they refer a youth in need of meds to some other health care level, and if so, which one? The authors state (line 327-328) that the absence of psychosocial support could lead to an increased prescribing of antidepressants. It could actually be the other way around; with increased access to psychosocial counselling, more youth would meet someone who could identify the need for pharmacological treatment.

Given the unclarity of the extent to which the included cohort represents the general population, think the title may be misleading. It is stated that this is a "population-based cohort study" - but if only part of the population is included in the study that must be clearly stated. 

The paper would improve by including perspectives from other countries in Europe and elsewhere, e.g Abbin-Karahagopian, Herta ea (2014, Eur J Clin Pharmacol), Noordam ea (2015, Eur J Clin Pharmacol), Lagerberg ea (2018, J European J of Child and Adolescent Psychiatry). 

The stated aims were to describe trends, estimate variation in uptake, and assess to what extent the NICE guidelines were adhered to. I am not sure what "variation in uptake" actually means in this study, especially not knowing to what extent this database covers the entire population. 

Please explain the term "Townsend deprivation quintiles".

*** Reviewer #4:

 This is an interesting and useful study on the trends and variation in the incidence and prevalence of antidepressant prescribing in children and young people in England. The study design, datasets, statistical methods and analyses, and presentation (tables ang figures) and interpretation of the results are mostly adequate. However, there are still a few important issues needing attention.

1) Strictly speaking, it's a cohort study. As it's not about incidence/prevalence in the population, it's better not to use 'population-based' in case mis-understanding and mis-interpretation.

2) In the conclusion of the abstract, it says 'the trends and variation in antidepressant prescribing found may reflect true differences in need and risk factors, access to diagnosis and services, prescribing behaviour, or young people's help-seeking behaviour'. However, apart from analyses of trends in incidence and prevalence of antidepressant prescriptions over time, there are no data or formal analyses on all those explanatory factors listed in the conclusion such as risk factors, access to services, prescribing behaviours and etc. It's good that the authors have done a nice job on the trends of prescription of antidepressant, but in the term of comprehensively explaining the trends, it seems a bit lacking solid evidence and analyses. Can authors find out the data within the cohorts on underlying conditions/co-morbidities, and their severity, related hospital care, and other risk factors, and carry out analyses to show the trends so that can offer some explanations on trends of prescriptions? 

3) Figure 2 and 3 are useful. However, are they adjusted incidence rates? If not, can we see the adjusted incidence rates in one way or another (comprehensively adjusted for everything)?

4) Table 3 on IRRs are mostly fine. However, when talking about fully adjusted IRRs, it became a bit limited as only adjusted for age, sex, deprivation, region, and ethnicity which are mostly demographics. The other important factors such as diagnosis, case-mix, hospital care, and other risk factor are not adjusted. If the data are not available, then at least should address this carefully in the limitations.

***

[LINK]

---

## [Decision Letter · Decision Letter 2]

3 Jun 2020

Dear Dr. Jack,

Thank you very much for re-submitting your manuscript "Incidence and prevalence of primary care antidepressant prescribing in children and young people in England, 1998-2017: a population-based cohort study" (PMEDICINE-D-20-00542R2) for consideration at PLOS Medicine.

I have discussed the paper with editorial colleagues and it was also seen again by three reviewers. I am pleased to tell you that, provided the remaining editorial and production issues are dealt with, we expect to be able to accept the paper for publication in the journal.

[LINK]

Please let me know if you have any questions. Otherwise, we look forward to receiving the revised manuscript shortly. 

Sincerely,

Richard Turner, PhD

rturner@plos.org

Requests from Editors:

In your data statement (in the submission form, to appear in the article metadata upon publication) we suggest substituting the following point of access: https://www.qresearch.org/information/information-for-researchers/. 

We suggest quoting 1-2 additional quantitative findings in your abstract - for example, the finding quoted at line 329 would seem of interest. 

Also, we would suggest adding a sentence, say, to the abstract to quote quantitative observations on the classes of antidepressants prescribed. 

Please adapt the abstract so that 95% CI are quoted along with incidence rate ratios and p values. 

In the sentence addressing study limitations at the end of the "methods and findings" subsection of your abstract, please quote one further limitation - possibilities could be that antidepressant use has not been measured, and the issue of unmeasured confounding. We would also suggest including these issues in the element of the discussion section summarizing study limitations. 

You mention that the study protocol has been published (line 173). Please highlight analyses that were not prespecified. 

Please refer to the attached STROBE checklist in your methods section (e.g., "See S1_STROBE"). 

Please move the ethics statement from the end of the ms to the methods section. 

Noting instances of "p<0.0001" in the tables, please quote exact p values or "p<0.001" throughout, unless there are specific statistical reasons to do otherwise. 

Please remove the information on funding, competing interests and data availability from the end of the ms - this information will appear in the article metadata (via the submission form). 

Please make that "PLoS ONE" in the reference list

Comments from Reviewers:

*** Reviewer #1: 

The authors have adequately addressed all of my comments.

*** Reviewer #3: 

Comments on ms PMEDICINE-D-20-00542R2

"Trends and variations in the incidence and prevalence of antidepressant prescribing in children and young people in England: a population-based cohort study

The manuscript has been improved in several ways.

The authors provide more information on the clinical practice in England with regard to CYP with mental health problems. 

They have also provided more information on the QResearch database and how they have tried to link the results to other data (i.e. the Townsend deprivation index).

The strength of this study is the size of the cohort and the long time period studied (which the authors themselves have identified, line 455-466). The identified trends are similar to what is seen in many other countries. The authors also do a good job in trying to relate their findings to other data of interest (demographic variations etc).

Since the aim is primarily descriptive, it is a weakness (also identified) that there is no data provided on what is prescribed from the specialist level, and lack of information on diagnoses/indications. It is stated that "In the UK, prescribing antidepressants to CYP should only be done after assessment and diagnosis by a child and adolescent psychiatrist or other expertise in child and adolescent mental health". To me, this seems as a quite strong recommendation and most likely, the results would be affected if specialist level data was provided alongside what is prescribed within on primary care level.

I agree with reviewer #4 that this is should be viewed as a cohort study; it could actually strengthen the argumentation. The authors provide support for the use of the QResearch database, and its representativity of the entire population. I accept that - but still we can assume that a lot of information on the issue in focus - prescription of antidepressants to CYP- are not included in this dataset. 

Given that the recommendation is that antidepressants should be described by specialists, the authors may claim that they with this impressive data set can show that this is not the case. They do have strong evidence that there is a large number of prescriptions of antidepressants being done within the primary care health services, with regional and other variations. Given that the specialist mental health services should be responsible for prescription of antidepressants one would assume that there should be very little (or no) prescriptions, or that the prescription patterns within primary care should be affected by change in policy or access to mental health services. The last question is also what the authors have tried to study when comparing regional variations in prescribing with other data on mental health resources.

One aim of the study was to assess to what extent the NICE guidelines for CYP are being adhered to. Is it possible to claim that based on these findings, guidelines are generally adhered to with regard to which antidepressants are prescribed, but not the way in which they are prescribed? I understand that GPs many times may need to deal with problems that "should" be handled on the specialist level, and that this pragmatic way of taking care of a medical problem may work out just fine and may be better than doing nothing. Still, these data provide a matrix for discussing the strengths and weaknesses in how youth get access to care.

I think the study has its merits, but primarily in the sense described above. It does not give a full picture of prescription patterns in England but rather a full picture of prescription patterns within primary care in England. The title should be adjusted accordingly.

*** Reviewer #4: 

I am satisfied with the authors' response. No further issues needing attention.

***

[LINK]

---

## [Editor Report · Decision Letter 3]

23 Jun 2020

Dear Dr Jack, 

On behalf of my colleagues and the academic editor, Dr. Clara Hellner, I am delighted to inform you that your manuscript entitled "Incidence and prevalence of primary care antidepressant prescribing in children and young people in England, 1998-2017: a population-based cohort study" (PMEDICINE-D-20-00542R3) has been accepted for publication in PLOS Medicine. 

PRODUCTION PROCESS

PRESS

PROFILE INFORMATION

Thank you again for submitting the manuscript to PLOS Medicine. We look forward to publishing it. 

Best wishes, 

Richard Turner, PhD

Senior Editor 

PLOS Medicine

plosmedicine.org